# Bitter Taste Receptor T2R14 and Autophagy Flux in Gingival Epithelial Cells

**DOI:** 10.3390/cells13060531

**Published:** 2024-03-17

**Authors:** Nisha Singh, Ben Ulmer, Manoj Reddy Medapati, Christine Zhang, Robert J. Schroth, Saeid Ghavami, Prashen Chelikani

**Affiliations:** 1Manitoba Chemosensory Biology (MCSB) Research Group, Department of Oral Biology, University of Manitoba, 780 Bannatyne Avenue, Winnipeg, MB R3E 0W4, Canada; nisha.singh@umanitoba.ca (N.S.); benjamin.ulmer@mail.mcgill.ca (B.U.); mmedapat@uwo.ca (M.R.M.); robert.schroth@umanitoba.ca (R.J.S.); 2Rady Faculty of Health Sciences, Dr. Gerald Niznick College of Dentistry, University of Manitoba, Winnipeg, MB R3T 2N2, Canada; 3University of Manitoba Flow Cytometry Core Facility, Rady Faculty of Health Sciences, University of Manitoba, Winnipeg, MB R3T 2N2, Canada; christine.zhang@umanitoba.ca; 4Rady Faculty of Health Sciences, Max Rady College of Medicine, University of Manitoba, Winnipeg, MB R3T 2N2, Canada; 5Research Institute of Oncology and Hematology, Cancer Care Manitoba, University of Manitoba, Winnipeg, MB R3E 0V9, Canada; 6Department of Human Anatomy and Cell Science, University of Manitoba College of Medicine, Winnipeg, MB R3E 0W2, Canada

**Keywords:** bitter taste receptor, acridine orange, serum starvation, autophagy flux inhibition, autophagosome, calcium, cariogenic bacteria

## Abstract

Macroautophagy (hereafter autophagy) is a lysosomal degradation pathway that functions in nutrient recycling and as a mechanism of innate immunity. Previously, we reported a novel host–bacteria interaction between cariogenic *S. mutans* and bitter taste receptor (T2R14) in gingival epithelial cells (GECs), leading to an innate immune response. Further, *S. mutans* might be using the host immune system to inhibit other Gram-positive bacteria, such as *S. aureus*. To determine whether these bacteria exploit the autophagic machinery of GEC, it is first necessary to evaluate the role of T2R14 in modulating autophagic flux. So far, the role of T2R14 in the regulation of autophagy is not well characterized. Therefore, in this study, for the first time, we report that T2R14 downregulates autophagy flux in GECs, and T2R14 knockout increases acidic vacuoles. However, the treatments of GEC WT with a T2R14 agonist and antagonist did not lead to a significant change in acidic vacuole formation. Transmission electron microscopy morphometric results also suggested an increased number of autophagic vesicles in T2R14-knockout GEC. Further, our results suggest that *S. mutans* competence stimulating peptide CSP-1 showed robust intracellular calcium release and this effect is both T2R14- and autophagy protein 7-dependent. In this study, we provide the first evidence that T2R14 modulates autophagy flux in GEC. The results of the current study could help in identifying the impact of T2R in regulation of the immuno-microenvironment of GEC and subsequently oral health.

## 1. Introduction

Autophagy is a lysosomal degradation pathway which plays a role in nutrient recycling and energy generation. It is also involved in the clearance of damaged proteins, organelles, and certain pathogens. Thus, it might be involved in part of the innate immune response [1,2]. The conventional process regulating the creation of autophagosomes relies on ATG7 enzyme activity, which triggers the activation of mammalian counterparts of ATG8 family proteins and the ATG12 protein [3]. This activation enables them to bind to lipid phosphatidylethanolamine (PE) and ATG5, respectively. However, recent research indicates that autophagy might also be governed by pathways that seem to function separately from ATG7 [4,5,6]. Previous studies have revealed that autophagy participates in odontoblast aging, tooth development, and many more physiological and pathological processes [7,8]. The involvement of many nutrient sensing G protein-coupled receptors (GPCRs) like umami and fat taste receptors within the autophagy phenomenon is well understood [9,10]. However, the involvement of bitter taste receptors (T2Rs) in the autophagy pathway is still unknown. To the best of our knowledge, only one study has shown a direct link between T2Rs and autophagy in airway smooth muscle (ASM) cells. This study showed that T2R agonists quinine and chloroquine alter mitochondrial function, induced cell death, and changed the number of autophagosomes via autophagy protein 5 (ATG5) and the coiled-coil myosin-like BCL2-interacting protein 1 (Beclin 1) pathway in ASM cells [11,12]. Chloroquine and quinine resulted in microtubule-associated light-chain protein (LC3β-II) accumulation, and this LC3β-II accumulation in ASM cells was surprisingly reduced in the presence of the autophagy flux inhibitor bafilomycin A1 (Baf A1). Thus, this finding suggests that T2R agonist-mediated cell death in human ASM cells is potentially regulated by changes in the autophagy pathway [11]. 

Oral gingival epithelial cells (GECs) are the first line of defense and produce antimicrobial peptides (AMPs) against various cariogenic bacteria. An invasion of non-phagocytic cells like epithelial, endothelial, and fibroblast cells is a common strategy of eluding the host immune system for many pathogens. These pathogens developed various survival mechanisms inside the host cells, and autophagy is one of them. Previous transcriptome analysis suggested a significant expression of T2R14 in GEC compared to other GPCRs and Toll-like receptors [13]. In a follow-up study, we suggested that T2R14 modulates Gram-positive bacterial internalization and survival in GECs [14]. The GECs infected with *Staphylococcus aureus* induced T2R14-dependent human β-defensin-2 (hBD-2) secretion, and T2R14 knockout affects the cytoskeletal reorganization, thereby inhibiting *S. aureus* internalization [14]. Since the internalization of *S. aureus* is T2R14-dependent, it is imperative to elucidate whether *S. aureus* and other cariogenic bacteria *S. mutans* use autophagic machinery for their survival and evading the host immune system. In order to determine whether such bacteria exploit the autophagic machinery of GEC, it is first necessary to evaluate the role of T2R14 in modulating autophagic flux. Autophagy is a fundamental cellular process relying on the formation of double-membrane vesicles, called autophagosomes, for the delivery of cytoplasmic proteins and organelles to the lysosomal degradation machinery for energy production and cellular homeostasis [15,16]. The number of autophagosomes is controlled via autophagy flux which is the outcome of autophagosome formation and degradation [17]. The initiation and nucleation of autophagosome formation is controlled by two major inductive protein complexes: the mammalian target of rapamycin (mTOR) complex 1 (mTORC1) and the Beclin 1 complex. The activity of the mTOR kinase suppresses autophagosome formation and acts as a negative regulator of autophagy. Rapamycin induces autophagic activity by inhibiting mTOR phosphorylation and thus preventing its kinase activity. Therefore, the regulation of the autophagic flux under certain physiological or pathological conditions (e.g., an imbalance of nutrient supply, oral pathogen invasion) is of particular interest in GECs. In this study, we provide evidence that T2R14 modulates autophagic flux in GECs, and that this phenomenon is ATG7-independent. Furthermore, our results also suggested that the activation of T2R14 with *S. mutans* competence-stimulating peptide (CSP-1) showed robust intracellular calcium release, and this effect is both T2R14- and ATG7-dependent in GEC.

Further, our result also suggests that the activation or inhibition of T2R14 activity upon agonist and antagonist treatments did not lead to any significant changes in the autophagy flux. 

## 2. Materials and Methods

### 2.1. Reagents Used in the Study

Acridine orange (CAS Number: 65-61-2), bafilomycin A1 (Cat#B1793-10UG), and rapamycin were purchased from Sigma-Aldrich Co., Oakville, ON, Canada. Synthetic *S. mutans* competence-stimulating peptide (CSP-1) and *S. aureus* auto-inducer peptide (AIP) of 98% purity were purchased from GenScript (Piscataway, NJ, USA). Keratinocyte growth medium-2 (KGM-2) for the OKF6 cell culture and Iscove’s modified Dulbecco’s medium (IMDM) for promyeloblast (HL-60) cells were purchased from promo cell (Heidelberg, Germany) and ATCC (Gaithersburg, MD, USA), respectively. The following antibodies were purchased: mouse monoclonal anti-β-actin (#A5441) from Sigma Aldrich (Oakville, ON, Canada), goat anti-rabbit IgG-HRP conjugate (#17-6515) from Bio-Rad (Mississauga, ON, Canada), goat anti-mouse IgG-HRP conjugate (#A-10668), and rabbit polyclonal anti-T2R14 (#OSR00161W) from Thermo Fischer Scientific, Carlsbad, CA, USA, and rabbit polyclonal ATG7 from cell signaling. Diphenhydramine (DPH) and 6-methoxyfalvanone were purchased from Sigma Aldrich (Oakville, ON, Canada).

### 2.2. Cell Line Used in the Study 

The oral keratinocyte cell line OKF6 was a kind gift from Dr. Gill Diamond, University of Florida [18,19]. The T2R14 KO and a non-targeting (MOCK) Alt^®^-R-control CRISPR- crRNA or MOCK OKF6 cells, here referred to as WT-mock, were generated using the CRISPR-Cas9 technique, as published in our previous study [13,20]. In total, 30–40 single clones obtained from serial dilutions in 96 plates were screened for mutations in the TAS2R14 gene using a T7E1 assay. We got two biallelic clones; of them, one was used in the current study and in our previous studies. The clone used in the current study had good cell viability and a healthy epithelial cell morphology. Further, for the reintroduction of TAS2R14 in T2R14 KO GEC, T2R14KO cells were plated in a 6-well plate, and TAS2R14 plasmid in the pcDNA 3.1 vector (3 μg) were transfected using lipofectamine 2000. After 24 h of cell transfection, cells were kept in hygromycin selection media (10 μg) for the expression of TAS2R14 in T2R14KO GEC, as described earlier [21]. The promyeloblast (HL-60) cells were cultured in IMDM supplemented with 20% FBS. The HL-60 cells were differentiated into neutrophils using previously established protocols [22].

### 2.3. Bacterial Strains Used in the Study 

The *S. aureus* (ATCC strain 6538, kind gift from Dr. Duan’s lab) and *S. mutans* strain UA159 were purchased from ATCC. The *S. aureus* strain was propagated in a Luria–Bertani (LB) broth at 37 °C, under constant agitation. The *S. mutans* strain was propagated in the brain heart infusion (BHI) broth at 37 °C and 5% CO_2_, under constant agitation. 

### 2.4. Co-Culture Assay of GECs

Briefly, for the co-culture assay, *S. aureus* and *S. mutans* were cultured overnight at 37 °C with shaking at 200 rpm in their respective culture media, and the GECs were infected with either *S. aureus* or *S. mutans* at a multiplicity of infection (MOI) of 50 and 100 for 16–18 h, respectively. Cell viability post-infection was assessed using the WST-1 assay [14]. After the treatments, the conditioned medium (CM) was collected and filtered using a 0.2 μm nylon filter. The filtered CM was used to determine the dHL-60 cell migration assay, as described previously [13]. 

### 2.5. Transmission Electron Microscopy Imaging of GECs 

The OKF6 WT and T2R14 KO (at least one million cells) were fixed with 3% gluteraldehyde in 0.1 M Sorensen’s buffer for 3 h. After fixation, the cells were suspended in a sucrose solution and stored at 4 °C for processing. The fixed cell suspensions were embedded into plastic resins and thin sections (90–100 nm) were placed on mesh copper grids. Finally, the copper grids were stained with osmium tetroxide and uranyl acetate [23]. The grids were imaged using a Philips CM10 microscope at a magnification of 10,500×. For evaluation of autophagy, cytosolic double-membrane vesicles were counted in 10 different fields with the same magnification. 

### 2.6. Acridine Orange Acidic Vacuole Assay 

We measured autophagy flux using our recently established protocol for GECs [24]. The autophagic flux was assessed using the Acridine Orange (AO) vacuolar assay for detecting acidic vesicular organelles (AVO), which are the indispensable markers of autophagy [24,25,26]. Acridine Orange staining induces green fluorescence in the cytosolic and nuclear fractions of cells, whereas when it becomes protonated, it emits intense red fluorescence upon fusion with the acidic environment, such as lysosomes. Thus, the ratio of red/green fluorescence intensity could be a suitable marker for AVO formation. GECs (25,000 cells) were seeded in 12-well plates under serum-starved conditions and treated for 72 h. After 72 h, cells were stained with AO (final concentration of 1 µg/mL) and incubated at 37 °C for 10 min in the dark. After washing once with PBS, the flow cytometry analysis was performed for evaluating the autophagy flux (red and green fluorescence) using a cytoflex LX flow cytometer (Brea, CA, USA) [24,26,27]. Further, the autophagy flux was evaluated in WT GECs under serum-starved conditions (48 h), and after that, treating the cells for 24 h with T2R14 agonist DPH (EC_50_ = 500 µM) and antagonist 6-MF (IC_50_ = 30 µM) using AO vacuolar assay. Treatments of GECs for 24 h did not lead to any significant changes in cell viability, as determined by the WST-1 assay. 

### 2.7. Immunoblot Analysis 

The protein expression level of ATG7 was quantified through immunoblotting. Cell lysates were prepared from control and ATG7 KD-specific cells. Protein estimation in the samples was performed using the Bio-Rad DC protein estimation kit. Each sample, containing 20 µg of protein, was resolved on a 10% SDS-PAGE and subsequently transferred onto nitrocellulose membranes. These membranes were blocked using 5% non-fat dry milk for one hour at room temperature, followed by incubation with specific primary antibodies overnight at 4 °C. The proteins were visualized by chemiluminescence, employing horseradish peroxidase (HRP)-conjugated secondary antibodies and enhanced chemiluminescence (ECL) substrate, with images captured on a ChemiDoc MP imaging system (Bio-rad, Mississauga, ON, Canada). The exposure times for chemiluminescent detection were set at 30 s for ATG7 and 2 s for the loading control, β-actin. 

### 2.8. Intracellular Calcium Mobilization Assay

Agonist-induced intracellular calcium mobilization assays were performed, as described previously [13,28]. WT, T2R14KO, ATG7 KD, and T2R14KO-ATG7KD cells (approx. 30,000/96 wells) were loaded with Fluo-4NW calcium dye, and intracellular calcium response was measured after AIP-1 (50 μM), CSP-1(50 μM) and CSP-1 plus Baf A1 treatment. Intracellular calcium changes were quantitatively measured using a Flex Station^®^ 3 MultiMode Microplate Reader (Molecular Devices, San Jose, CA, USA), with results expressed as alterations in relative fluorescence units (RFUs). Changes in intracellular concentration were recorded using the Flex Station^®^ 3 MultiMode Microplate Reader, with data represented as the change in relative fluorescence unit (RFUs). 

### 2.9. Immune Cell (dHL-60) Migration

The migration of dHL-60 cells is a standard model system for monitoring the neutrophil migration assay in the presence of chemoattractant in vitro [29]. Briefly, dHL 60 cells were seeded on the top chamber of the CIM plate at a density of 4 × 10^5^ cells/well in HBSS containing 0.5% BSA. The bottom chamber was loaded with a conditioned medium (CM) obtained from with or without *S. aureus* and *S. mutans* treatments in WT, T2R14KO, ATG7KD and T2R14KO-ATG7KD cells on the migration of differentiated HL-60 cells (dHL-60). The changes in cellular impedance or cell index (CI) were measured every 5 min over 24 h using RTCA software version 2.0 [13].

### 2.10. Knockdown of ATG7

Human GECs were initially placed in 12-well plates at a density of 5 × 10^4^ cells per well and cultured in a KGM-2 medium with growth supplements for 24 h. When the cells reached approximately 40% confluency, they were exposed to 10 μg/mL of polybrene (Santa Cruz; sc-134,220) in the KGM-2 basal medium for 1 h. Following this, the cells were transfected with shRNA lentiviral particle encoding for shRNA ATG7, and scrambled control which also carried the puromycin resistance marker (Santa Cruz; sc-41447-V). The transfection was conducted at 3 and 6 multiplicities of infections (MOI) for 12 h, after which the medium was replenished for a 24 h recovery period. To identify the cells that successfully incorporated the shRNA plasmids, they were grown in a medium containing puromycin dihydrochloride (4 μg/mL) (Santa Cruz; sc-108071). The Western blot technique was then used to examine the ATG7 status of both the shRNA-transfected cells and the scramble cells in isolated stable clones [30].

### 2.11. Statistical Analysis

All experimental results are presented as the mean ± standard error of the mean (SEM) from at least three independent experiments. GraphPad PRISM v9.0 (GraphPad Software, San Diego, CA, USA) was used to perform statistical significance analyses. A one-way analysis of variance (ANOVA) was applied to compare more than three groups. Data are presented as the mean ± standard error of the mean (SEM); * *p* < 0.05, ** *p* < 0.01, *** *p* < 0.001, **** *p* < 0.0001, as indicated.

## 3. Results

### 3.1. T2R14-Dependent Autophagy Flux in GEC

Our previous studies suggest T2R14 is the primary driver of innate immune responses to bacterial infection in GECs [13,14]. As previously shown, autophagy could be involved in both innate immunity [31] and bacterial infection [32]; therefore, we evaluated the effect of T2R14 expression and activity in autophagy flux in GECs, with oral keratinocyte cells, OKF6 Wt (WT), and OKF6 T2R14 KO (T2R14KO) cells being used as the model system. The autophagic flux was assessed using the Acridine Orange vacuolar assay for detecting acidic vesicular organelles (AVOs), which are the indispensable markers of autophagy [25,26]. Both cell lines were serum-starved for 72 h to induce autophagy [17]. The cells were also treated with an autophagy flux inhibitor, Bafilomycin A1 [Baf A1 (10 nM)], and an autophagy flux inducer, rapamycin [Rapa (1000 nM)], for 72 h. The treatment of GECs with Baf A1 and Rapa for 72 h did not lead to any significant change in cell viability (Figure 1A) and flow cytometry analysis of serum-starved GECs did not result in significant cell death. After 72 h, cells were stained with Acridine Orange (final concentration of 1 µg/mL) and incubated at 37 °C for 10 min in the dark. After washing once with PBS, the flow cytometry analysis was performed for evaluating the autophagy flux (Red and Green fluorescence) using the cytoflex flow cytometer [26]. Acridine Orange staining induces green fluorescence in the cytosolic and nuclear fractions of cells, whereas when it becomes protonated, it emits red intense fluorescence upon fusion with the acidic environment, such as lysosomes. Our results suggested that serum starvation in T2R14KO cells leads to a significant increase in red fluorescence compared to WT (Figure 1B), suggesting more acidic vacuoles in T2R14KO cells. Further, no change in the green fluorescence was observed (Figure 1C). Finally, the red/green fluorescence ratio was also significantly increased under T2R14KO serum-starved conditions compared to the WT serum-starved group (Figure 1D). Further, the treatments of WT and T2R14KO cells with autophagy flux inhibitor Baf A1 leads to the complete loss of red fluorescence (Figure 1E). Baf A1 treatments inhibits vacuolar H(+)-ATPase (V-ATPase), which results in the inability of the lysosome to acidify; thus, no red fluorescence was observed. As expected, autophagy flux inducer Rapa leads to an increase in red fluorescence (Figure 1E). 

Next, the TAS2R14 gene was reintroduced in T2R14KO GECs for rescuing the effect of T2R14-mediated autophagy flux. Our results suggested that T2R14KO GECs transfected with TAS2R14 were less viable (Appendix A) and grow slower than the T2R14KO GEC. Thus, we could not evaluate the T2R14 rescue effect on autophagy flux in these T2R14KO GECs.

Next, we used the T2R14 agonist diphenhydramine (DPH) to measure the autophagy flux in WT GECs under serum-starved conditions [13,33]. Using 500 µM (EC_50_) DPH did not lead to any change in cell viability with a 24 h treatment. Upon treatment with T2R14 agonist DPH at 500 µM under serum-starved conditions, WT GECs had non-significant changes in the number of acidic vacuoles (ratio of red/green) relative to the unstimulated cells. The results suggested that after activation of T2R14 alone, there was no significant change in the acidic vacuoles (Appendix A). 

To corroborate the data obtained using Acridine Orange staining of AVOs, transmission electron microscopy (TEM) was used for WT and T2R14 KO GECs. The TEM morphometric results suggested an increased number of autophagic vesicles (autophagosome and phagophore) in T2R14KO relative to the WT-mock (Figure 1F–H). Autophagic vacuoles were classified as autophagosomes when they met the following criteria: (1) double membrane (complete or at least partially visible), and (2) absence of the ribosomes attached to the cytosolic side of the membrane [34].

### 3.2. ATG7-Independent Autophagy Flux in GECs 

ATG7 is important for the formation of autophagic vesicles and is directly involved in the processing of LC3-I into LC3-II [17,35]. To check the involvement of ATG7 in autophagosome formation in GECs, ATG7 was knocked down in both WT and T2R14KO GECs using shRNA (Figure 2A). The loss of ATG7 in GECs resulted in non-significant changes in the red fluorescence or ratio of red/green fluorescence (Figure 2B,D). However, we did not observe significant changes in red and red/green fluorescence after Acridine Orange staining in T2R14KO-ATG7KD compared to WT-sc shRNA under serum-starved conditions (Figure 2B–E). This could be due to the ATG7-independent autophagy pathway in GECs. Previous studies suggested that autophagosome formation and autophagy process are independent of ATG7 in different cell types [5,36].

### 3.3. Intracellular Calcium Mobilization Assay of GECs

There have been several reports suggesting the involvement of calcium in the autophagy pathway [37,38]. Calcium is known to act as both a promoter and inhibitor of autophagy under different circumstances. It is well-known that the activation of T2Rs leads to intracellular calcium release via T2Rs-Gβγ–PLCβ-IP3-Ca^2+^ [13,39]. To elucidate whether changes in autophagy flux could affect the intracellular mobilization of calcium ion via activation of endogenous T2R14, GECs were stimulated with Baf A1 (10 nM) and Rapa (1000 nM) and intracellular calcium release were measured. Baf A1 leads to no significant changes in intracellular calcium release in ATG7 KD or T2R14KO cells compared to WT-treated cells (Figure 3A). Our previous findings suggested that in GECs, competence-stimulating peptide 1 (CSP-1) from cariogenic bacteria *S. mutans* exhibited T2R14-dependent intracellular calcium release [13]. We also demonstrated the T2R14-dependent *S. aureus* bacterial internalization and release of antimicrobial peptide (HBD2) in GECs [14]. However, the effect of autoinducing peptide 1 (AIP-1) from *S. aureus* and CSP-1 from *S. mutans* on intracellular calcium release in ATG7KD cells was not known. ATG7KD and T2R14KO GECs showed a lower calcium release compared to the WT upon CSP-1 treatment (Figure 3B). Next, the combined effect of Baf A1 and CSP-1 on calcium signaling was analyzed. The results suggested that there was no change in intracellular calcium release upon the co-treatment of GECs with Baf A1 and CSP-1 compared to CSP-1 treatment alone (Figure 3C). This suggests that autophagy flux does not impact T2R14-dependent calcium mobilization in GECs.

### 3.4. Immune Cell (dHL-60) Migration

Neutrophils are the first immune cells recruited to the site of gingival tissue inflammation upon bacterial infection. Our previous results suggested that the treatment of GECs with synthetic CSP-1 led to an increased secretion of the antimicrobial peptides (AMPs), chemokine interleukin 8 (IL-8) and human beta defensin 2 (HBD-2), in a T2R14-dependant manner [13]. To investigate whether the oral bacteria-infected GECs recruit immune cells that are T2R14- and/or ATG7-dependent, we tested the effect of a conditioned medium (CM) obtained from *S. aureus-* and *S. mutans*-treated GECs on the migration of neutrophils like cell line-differentiated HL-60 cells (dHL-60). The treatment of dHL-60 cells with the aforementioned conditioned media led to no changes in dHL-60 cells, which is consistent with our previous report [13]. Previously, we observed a significant decrease in dHL-60 migration upon treatment with conditioned media from T2R14KO cells primed with synthetic CSP-1 compared to mock cells. We did not observe a similar effect on dHL-60 migration when treated with conditioned media from either *S. aureus* or *S. mutans* (Figure 4A,B). This could be due to multiple reasons, including the low levels of peptides secreted by the microbes in our assay system, internalization of microbes by GECs, and the differences in post-translational modification of the peptides secreted by the microbes versus synthesized peptides. Further, the treatment of dHL-60 cells with BafA1 and Rapa were performed. The results suggested that there was no effect of these compounds on the migration of neutrophil-like cells (Figure 4C).

## 4. Discussion

Autophagy is a physiological compensation process involved in the maintenance of cell homeostasis via removing misfolded proteins and damaged organelles. It is also involved in the regulation of host cell response against microbial infection. However, the mechanism by which these oral pathogens utilize autophagy as a survival mechanism and elude the host immune system is not well understood. In this study, we characterized the role of T2R14 in modulating autophagy flux in GECs. On the basis of Acridine Orange staining of AVOs (Figure 1), the presence of T2R14 in GECs increases the number of acidic active lysosomes. This could be due to an effect of T2R14 on the activity of lysosomal enzymes, or an effect of T2R14 on the intracellular pH of lysosomes. Further study is required to confirm the effects of T2R14 on the lysosomal activity and the intracellular pH of lysosomes. Our results also suggested that the modulation of T2R14 activity by a T2R14 agonist and antagonist did not lead to any significant change in acidic active lysosomes under serum-starved conditions (Appendix A).

Both *ATG5* and *ATG7* are believed to be indispensable for the autophagy pathway [40]. Thus, in the present study, we knocked out the *ATG7* gene for performing the autophagy flux. Our results (Figure 2) suggested that the loss of ATG7 in GECs resulted in non-significant changes in the red fluorescence or ratio of red/green fluorescence. This could be due to the ATG7-independent autophagy pathway in GECs. Previous studies also suggested that autophagosome formation and autophagy process is independent of ATG7 in different cell types [5,36]. The presence of the ATG5/ATG7-independent, alternative autophagy pathway has been confirmed in a variety of cells, which includes fibroblasts and preadipocytes obtained from *Atg5*^flox/flox^ mice [41], thymocytes obtained from an *Atg5*^−/−^ mouse embryo [5], mouse pancreatic β-cells [42], and induced pluripotent stem cells (iPSCs) [41]. 

Previous reports suggested the involvement of calcium in the autophagy pathway [37]. Calcium is known to act as both a promoter and inhibitor of autophagy under different circumstances. It is well established that the activation of T2Rs leads to intracellular calcium release [13,39]. Our results suggested there was no change in intracellular calcium release upon the co-treatment of GECs with Baf A1 and CSP-1 compared to CSP-1 alone (Figure 3C). This suggests that the T2R14-mediated calcium release was not involved in T2R14-dependent autophagy flux in GECs. Studies suggested that intracellular calcium is not always associated with an increase or decrease in autophagy flux [38,43]. In general, the final effect of calcium on autophagy depends on the spatiotemporal characteristics and the amplitude of calcium signals, as well as cell growth conditions (e.g., nutrient and growth factor availability) and type of autophagy [44]. Recent investigations have shown the potential impact of ATG7-dependet autophagy on calcium flux and ER. Ultraviolet (UV) irradiation triggers autophagy in the epidermis, and the autophagy gene ATG7 has been found to promote UV-induced inflammation and skin tumorigenesis [45]. ATG7 controls UV-induced cytokine expression and secretion, and enhances the expression of Ptgs2/Cox-2 through both a cell-autonomous mechanism involving CREB1/CREB and a non-cell-autonomous mechanism mediated by IL1B/IL1β. The addition of PGE2 exacerbates UV-induced skin inflammation and tumorigenesis, reversing the epidermal phenotype observed in mice with ATG7 deletion in keratinocytes. Similarly, the knockout of ATG5 in human keratinocytes inhibited the UVB-induced expression of PTGS2 and cytokines. Moreover, the loss of ATG7 resulted in an increased activation of the AMPK pathway and phosphorylation of CRTC1, along with the accumulation of the endoplasmic reticulum (ER) and reduction in ER stress. Treatment with thapsigargin, which induces ER stress and inhibits calcium influx into the ER, reversed the inflammation and tumorigenesis phenotype in mice with an epidermal ATG7 deletion [45].

The infiltration of immune cells such as neutrophils and macrophages led to gingival tissue inflammation [46]. In recent years, autophagy machinery has emerged as a central regulator of innate immune functions, cytokine production, immune cell differentiations, and pathogen clearance [47]. Neutrophils are the first line of immune cells recruited to the site of gingival tissue inflammation upon bacterial infection. Thus, the effect of a conditioned medium obtained from *S. aureus-* and *S. mutans*-treated GECs on the neutrophil-like cell line dHL-60 was investigated. The treatment of dHL-60 cells with conditioned media led to no changes in dHL-60 cells, which is consistent with our previous report [13]. Previously, we observed a significant decrease in dHL-60 migration upon treatment with conditioned media from T2R14KO cells primed with synthetic CSP-1 compared to mock cells. We did not observe a similar effect on dHL-60 migration when treated with conditioned media from either *S. aureus* or *S. mutans* (Figure 4A,B). It is possible that *S. aureus* and *S. mutans*, once internalized inside the GECs, are unable to elicit sufficient cytokine production involved in the neutrophil migration. Since T2R14 differentially modulates the internalization of *S. aureus* and *S. mutans* inside GECs, it is imperative to analyze how these Gram-positive bacteria manipulate host cells for their survival and their effects on oral innate immunity. Further studies to investigate whether cariogenic and non-cariogenic bacteria use similar autophagy processes to form their replicative niche in GECs are warranted. 

The mammalian target of rapamycin complex 1 (mTORC1) plays a crucial role in regulating cell growth, metabolism, and autophagy. While considerable research has focused on pathways that activate mTORC1, such as growth factors and amino acids, less is known about the signaling mechanisms that directly inhibit its activity. Recent investigation demonstrated that G protein-coupled receptors (GPCRs) linked to Gαs proteins elevate cyclic adenosine 3′5′ monophosphate (cAMP) levels, activating protein kinase A (PKA) and subsequently inhibiting mTORC1 [48]. Mechanistically, PKA phosphorylates the mTORC1 component Raptor at Ser 791, resulting in reduced mTORC1 activity. This inhibition of mTORC1 persists even upon PKA activation in cells where Raptor Ser 791 is mutated to Ala, indicating the importance of this phosphorylation site [48]. 

Previous studies have suggested that the autophagosome or autophagic vesicle formation involves both non-canonical and canonical pathways [49]. Future studies evaluating the role of lysosomal small Rho GTPase, mTOR, and Beclin 1 complexes in the regulation of T2R-mediated autophagy in GECs and other cell types can be pursued. It is far beyond the aims of the present study to investigate the complete signaling pathway involved in the autophagic machinery in the context of oral innate immunity. On the other hand, autophagy could be a master regulator of chemokine and cytokine secretion from mammalian cells through secretory autophagy and unfolded protein response [50,51]. Proteins destined for secretion through traditional pathways typically possess an N-terminal signal peptide, facilitating their transport to the endoplasmic reticulum and Golgi apparatus for eventual secretion. However, certain cytosolic proteins lack these signal peptides and thus require alternative, unconventional, or “non-canonical” secretion mechanisms [52]. One such unconventional pathway is termed secretory autophagy (SA), closely associated with the autophagy process. SA relies on ATG proteins that regulate autophagosome formation, the key organelle involved. Unlike canonical macroautophagy, which involves autophagosome–lysosome fusion for degradation, SA bypasses this degradation step, enabling secretion. ATG5 and other autophagy-related factors, like BCN1, are also activated in this secretory pathway. SA has emerged as a significant mechanism for the unconventional secretion of various cytosolic proteins critical for biological processes, including cytokines and chemokines that might regulate inflammation [52]. Additionally, SA may aid in the release of proteins prone to aggregation, as it intersects with autophagosome biogenesis machinery. Notably, SA operates within the autophagic pathway, and both secretory autophagy and canonical degradative autophagy are intricately linked and tightly regulated processes, interacting through complex molecular mechanisms [53]. Future investigations addressing if T2Rs are involved in the regulation of the secretion and processing of GEC-originated chemokines and cytokines via SA and unfolded protein response are very much needed. 

## Figures and Tables

**Figure 1 cells-13-00531-f001:**
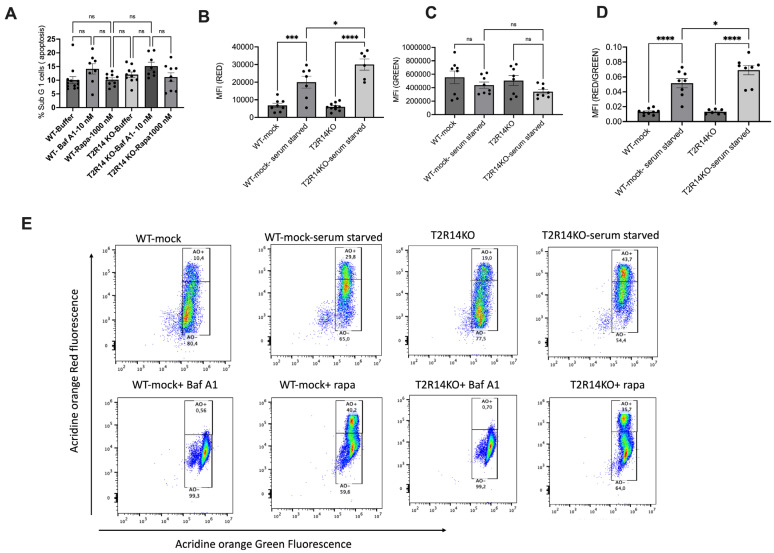
T2R14-dependent autophagy flux in gingival epithelial cells (GECs). (**A**) Effect of Baf A1 and Rapa on GEC viability. Flow cytometry cell-based assay suggested the effect of Baf A1(10 nM) and Rapa (1000 nM) on the cell growth in both WT-mock and T2R14KO cells over 72 h. The bar plots represent the SEM of three independent experiments performed in triplicates. (**B**–**D**) Acridine Orange staining of AVOs for detecting autophagy flux in GECs. Flow cytometry evaluation of the red, green, and red/green fluorescence ratio for detecting the autophagy flux in WT-mock and T2R14KO in GECs under serum-starved conditions and their respective normal growth condition. The bar graphs were generated using GraphPad PRISM 9.0. Statistical significance was calculated using a one-way ANOVA with Bonferroni’s post hoc test * *p* < 0.05, *** *p* < 0.001, **** *p* < 0.0001 and ns denotes non-significant. The data represent the SEM of 3–4 independent experiments. (**E**) Acridine Orange staining of AVOs for detecting autophagy flux in GECs. Representative raw traces showing the flow cytometric detection of red and green fluorescence in Acridine Orange-stained GECs (WT-mock and T2R14 KO) that were serum-starved or treated with Baf A1 (10 nM) and Rapa (1000 nM) for 72 h for detecting the autophagy flux. (**F**–**H**) Transmission electron microscopy (TEM) evaluation of autophagic vacuoles in WT-mock and T2R14KO cells. The white-headed arrow indicates the phagophore (incomplete autophagosome) and autophagosome double-membrane structure (Insets) in respective GECs (**F**,**G**). Scale = 2 microns at magnification of 10,500×. (**H**) Quantification of autophagosome based on the morphology (double membrane) in WT-mock and T2R14 KO cells. The results represent the mean value of autophagic vesicles per cell.

**Figure 2 cells-13-00531-f002:**
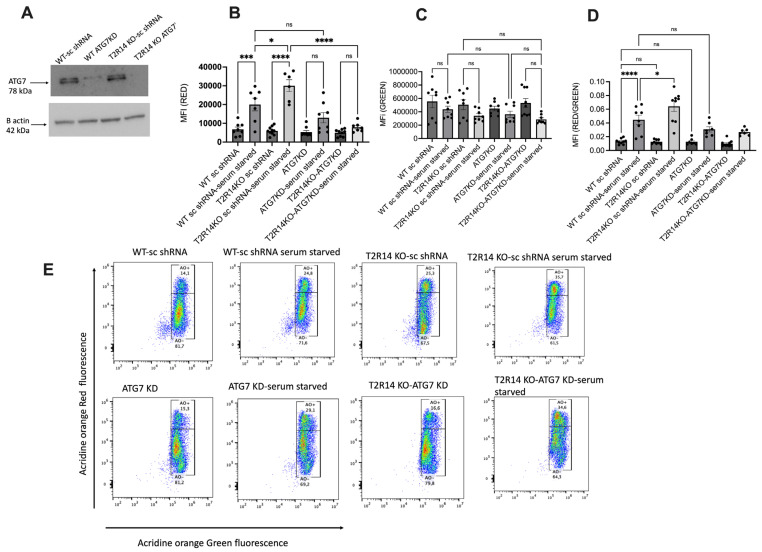
ATG7-independent autophagy flux in gingival epithelial cells (GEC). (**A**) Western blot analysis of ATG7 knockout in WT and T2R14KO GECs. ATG7 KD in WT and T2R14 KO GECs was performed using an shRNA lentiviral approach. WT denoted WT-sc shRNA, and T2R14KO denoted T2R14 KO sc shRNA. (**B**–**D**) Acridine Orange staining of AVOs for detecting autophagy flux in GECs. Flow cytometry evaluation of the red, green, and red/green fluorescence ratio for detecting the autophagy flux in ATG7KD and T2R14KO-ATG7KD in GECs under serum-starved conditions and their respective normal growth condition. The bar graphs were generated using GraphPad PRISM 9.0. Statistical significance was calculated using a one-way ANOVA with Bonferroni’s post hoc test * *p* < 0.05, *** *p* < 0.001, **** *p* < 0.0001 and ns denotes non-significant. The data represent the SEM of 3–4 independent experiments. (**E**) Acridine Orange staining of AVOs for detecting autophagy flux in GECs. Representative raw traces showing the flow cytometric detection of red and green fluorescence in Acridine Orange-stained GECs (ATG7KD and T2R14-ATG7KD) that were serum-starved for 72 h for detecting the autophagy flux.

**Figure 3 cells-13-00531-f003:**
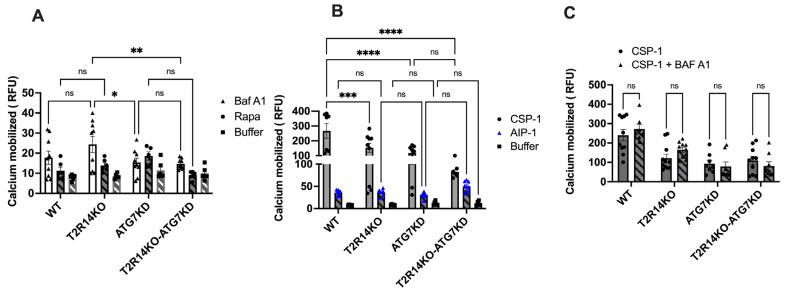
Intracellular calcium mobilization assay in GECs. (**A**) Pharmacological characterization of autophagy flux inhibitor, Baf A1 (10 nM) and autophagy inducer Rapa (1000 nM) on the calcium mobilization in GECs. WT, T2R14KO, ATG7 KD, and T2R14KO-ATG7KD cells (approx. 30,000/96 wells) were loaded with Fluo-4NW calcium dye and the intracellular calcium response was measured after Baf A1 and Rapa treatment. The data shown are the mean ± SEM of three independent experiments performed in triplicate. (**B**) Effect of *S. aureus* autoinducer peptide (AIP-1) and *S. mutans* competence-stimulating peptide (CSP-1) on calcium mobilization in GECs. Intracellular calcium release measurement after the application of autoinducer peptides (AIP from *S. aureus* and CSP-1 from *S. mutans* (50 µM each) in GECs. Results are from three independent experiments performed in triplicate. (**C**) Combined effect of Baf A1 and CSP-1 on calcium signaling in GECs. WT, T2R14KO, ATG7 KD, and T2R14KO-ATG7KD cells (approx. 30,000/96 wells) were loaded with Fluo-4NW calcium dye and the intracellular calcium response was measured after CSP-1 and CSP-1 plus Baf A1 treatment. The data shown are the mean ± SEM of three independent experiments performed in triplicate. Statistical significance was calculated using a one-way ANOVA with Bonferroni’s post -hoc test * *p* < 0.05, ** *p* < 0.01, *** *p* < 0.001, **** *p* < 0.0001 and ns denotes non-significant.

**Figure 4 cells-13-00531-f004:**
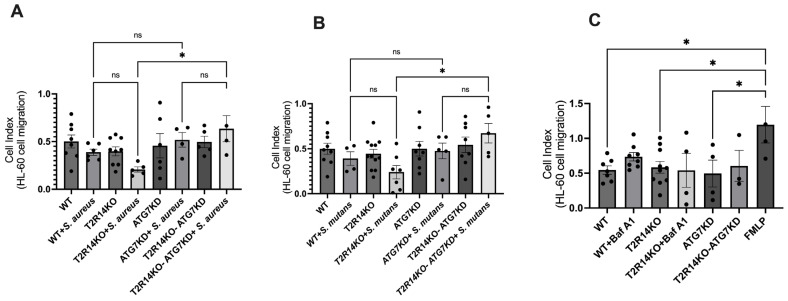
Immune cell (dHL-60) migration. (**A**,**B**) Conditioned medium (CM) obtained from with treatments in WT, T2R14KO, ATG7KD, and T2R14KO-ATG7KD cells with or without *S. aureus* and *S. mutans* on the migration of differentiated HL-60 cells (dHL-60). The changes in cellular impedance or cell index (CI) were measured every 5 min over 24 h using RTCA software. The bar graph shows change in the migration of dHL-60 cells over a period of 24 h. The bar plots represent the SEM of three independent experiments and were generated using GraphPad PRISM 9.0. Statistical significance was calculated using a one-way ANOVA with Bonferroni’s post hoc test * *p* < 0.05, and ns denotes non-significant. (**C**) Effect of conditioned medium (CM) obtained from (with or without BafA1 (10 nM) treatment) WT, T2R14KO, and ATG7KD and T2R14KO-ATG7KD cells on the migration of differentiated HL-60 cells (dHL-60). The changes in cellular impedance or cell index (CI) were measured every 5 min over 24 h using RTCA software. FMLP was included as a positive control for neutrophil chemoattractant. The bar plots represent the SEM of 2–3 independent experiments performed in triplicates. * *p* < 0.05 showed statistical significance.

## Data Availability

The data that support the findings of this study are available on reasonable request from the corresponding author.

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
