# Peer review of "Bitter Taste Receptor T2R14 and Autophagy Flux in Gingival Epithelial Cells"

_cells, 2024, doi:10.3390/cells13060531_

Round 1

Reviewer 1 Report

Comments and Suggestions for Authors The authors have provided a very novel and innovative finding on the mechanisms of bitter taste receptor (T2R) induced autophagy in Gingival Epithelial Cells (GEC) and its impact in regulation of immune response to bacterial infections. They have used T2R KO and Atg7 KD for their investigations and also provided AO and TEM evaluation of autophagy.    Major Comments: Introduction: In the introduction it will be more informative for readers if the authors add more information about Atg7-dependent autophagy and also regulatory effect of autophagy in the immune response of epithelial cells.    Methods: Detailed explanation about the confluent of the cells (starting, final time point) for measuring autophagy. Also provide more detail on evaluation of autophagy using TEM and AO. It will help the reproducibility of the work when other researchers will use the method in the paper.    Results: 1- I recommend the authors use the "Autophagometer" article and explain further the evaluation of flux. This is one of the most tricky evaluations and really needs further discussion.    2- The authors need better explanation of the source of changes of intracellular evaluation of calcium. Is it mitochondrial, ER? Could it affect the ER?   3- The missing point of the article is that the author provides evidence that changing of T2R and Atg7 expression together may affect immune response of GEC. Do they have any evidence? The authors have previously shown it in their T2R KO model. How about the effect of flux inhibition in GEC WT and GEC T2RKO?   Discussion: It is expected the authors have deeper discussion about the potential cross talk of GPCR mediated regulation autophagy flux and provide any related literature.  Also, discussion about the potential impact of Atg7 in calcium mobilization and its impact in ER and UPR induction and changing the flux of autophagy through the UPR autophagy flux cross talk.     Minor: The graphical abstract needs more work with further details on their findings. 

Author Response

The authors have provided a very novel and innovative finding on the mechanisms of bitter taste receptor (T2R) induced autophagy in Gingival Epithelial Cells (GEC) and its impact in regulation of immune response to bacterial infections. They have used T2R KO and Atg7 KD for their investigations and also provided AO and TEM evaluation of autophagy.   

Major Comments:

Introduction: In the introduction it will be more informative for readers if the authors add more information about Atg7-dependent autophagy and also regulatory effect of autophagy in the immune response of epithelial cells. 

Response: Thank you as suggested information is now added in the text (Introduction, page 1: 43-47 (Atg7-dependet autophagy), future direction page 11 and 12, lines (417-431, 468-484) (Secretory autophagy).

Methods: Detailed explanation about the confluent of the cells (starting, final time point) for measuring autophagy. Also provide more detail on evaluation of autophagy using TEM and AO. It will help the reproducibility of the work when other researchers will use the method in the paper. 

Response: As the review might acknowledge, the use of TEM and AO in autophagy studies is well accepted. Our recent publication (1), reference 25, described in detail methods pertaining for Acridine Orange staining for the evaluation of autophagy flux. Also, more detail about TEM is included in the methods (Lines 147-148)

  Results: 1- I recommend the authors use the "Autophagometer" article and explain further the evaluation of flux. This is one of the most tricky evaluations and really needs further discussion. 

Response: The authors appreciate the careful concern of the respected reviewer. We have recently published a separate method paper in Springer protocols (in-press, reference 25) and carefully addressed the autophagy flux measurement in GECs using immunoblotting in presence and absence of flux inhibition and induction (Baf-A1 and Rapa) based on the autophagometer paper  ((1) in press, reference 25). AO method is a faster method to detect autophagy and is more convenient while using primary cells lines (like our model) which grow slower.

2- The authors need better explanation of the source of changes of intracellular evaluation of calcium. Is it mitochondrial, ER? Could it affect the ER?  

Response: It is well known that activation of T2Rs leads to intracellular calcium release via T2Rs-Gβγ –PLCβ-IP3-Ca2+ (2, 3) (references 14 and 41)In the T2R field, it remains to be characterized  whether the source of intracellular calcium release upon T2R activation is mitochondrial or ER. The calcium release shown in figure 3 is total cytosolic intracellular calcium mobilized upon the stimulus of GECs with different compounds.

3- The missing point of the article is that the author provides evidence that changing of T2R and Atg7 expression together may affect immune response of GEC. Do they have any evidence? The authors have previously shown it in their T2R KO model. How about the effect of flux inhibition in GEC WT and GEC T2RKO?  

Response: In our previous work we have shown T2R14 KO in GECs influences innate immune responses against cariogenic bacteria (3, 4), we have not tested yet both T2R14KO and ATG7KD in innate immune responses in GEC. Something we hope to pursue in future.

Discussion: It is expected the authors have deeper discussion about the potential cross talk of GPCR mediated regulation autophagy flux and provide any related literature.  Also, discussion about the potential impact of Atg7 in calcium mobilization and its impact in ER and UPR induction and changing the flux of autophagy through the UPR autophagy flux cross talk.    

Response: As suggested, we have included this in the discussion lines 417-431 and lines 451-460. GPCR and autophagy (lines 451-460).

Minor: The graphical abstract needs more work with further details on their findings. 

Response: While the authors appreciate the respected reviewer suggestion, we prefer to keep the graphical abstract as simple as possible, so it captures the main findings and more understandable for lay reader of the paper.

  1. Singh N, Ghavami S, Chelikani P. Characterization Of Bitter Taste Receptor Dependent Autophagy in Oral Epithelial Cells. bioRxiv. 2024:2024.02.02.578576.
  2. Zhou YW, Sun J, Wang Y, Chen CP, Tao T, Ma M, et al. Tas2R activation relaxes airway smooth muscle by release of Gα. Proc Natl Acad Sci U S A. 2022;119(26):e2121513119.
  3. Medapati MR, Singh N, Bhagirath AY, Duan K, Triggs-Raine B, Batista EL, Jr., et al. Bitter taste receptor T2R14 detects quorum sensing molecules from cariogenic Streptococcus mutans and mediates innate immune responses in gingival epithelial cells. FASEB J. 2021;35(3):e21375.
  4. Medapati MR, Bhagirath AY, Singh N, Schroth RJ, Bhullar RP, Duan K, et al. Bitter Taste Receptor T2R14 Modulates Gram-Positive Bacterial Internalization and Survival in Gingival Epithelial Cells. Int J Mol Sci. 2021;22(18).
  5. Carey RM, Palmer JN, Adappa ND, Lee RJ. Loss of CFTR function is associated with reduced bitter taste receptor-stimulated nitric oxide innate immune responses in nasal epithelial cells and macrophages. Front Immunol. 2023;14:1096242.
  6. Babatunde KA, Wang X, Hopke A, Lannes N, Mantel PY, Irimia D. Chemotaxis and swarming in differentiated HL-60 neutrophil-like cells. Sci Rep. 2021;11(1):778.

Reviewer 2 Report

Comments and Suggestions for Authors

Review of manuscript cells-2853868

The paper by Singh et al, entitled “Bitter taste receptor T2R14 and autophagy flux in gingival epithelial cells” is a continuation of an investigation, by the same authors, on the involvement of bitter taste receptors, namely Tas2R14 (T2R14), on gingival epithelial cells-mediated innate immunity (references 10 and 11 in current manuscript draft). In the current study, the authors conduct a series of experiments aimed at uncovering potential T2R14-dependent modulation of autophagy flux in an oral keratinocyte cell line (OKF6), revealing some the molecular/cellular mechanisms governing autophagy in these cells, and the impact on autophagy flux modulation (with the use of pharmacological blockers), and T2R14, on innate immune response, through measurement of migration of neutrophil-like cells, using conditioned media derived from bacteria-infected OKF6 cells.

Unfortunately, the paper in its present form does not address the some of the authors’ questions precisely, and, in some instances, the data interpretation is inconsistent. Furthermore, some conclusions are not fully supported. In one of the last sentences of the introduction (lines 80 and 81), the authors state that they provide evidence that T2R14 modulate autophagic flux in GECs. Not enough data is provided to support this conclusion, in my opinion.

Point #1:

In the first series of experiments, the authors monitor autophagy flux in OKF6 cells, and a derived cell line where T2R14 was knocked out using CRISPR (T2R14KO). Figures 1B and 1D show a modest increase in the numbers of acidic vacuoles in the T2R14KO cells vs WT cells. Transmission electron microscopy data shown in Figures 1F-H seem to confirm this data. However:

1.1.        How can we exclude just a clonal effect of the T2R14KO GECs? The authors do not describe how the knockout cell line was generated but rather point to reference #10 as the source of information. In this refenced paper, CRISPR is used to generate T2R14 knock down OKF6 cells, and single clones are selected. So, what if the data provided in Figure 1 of the current paper is just the result of a specific OKF6 clone, exhibiting more acidic vacuoles upon starvation, while also having no expression of T2R14? If one re-introduces T2R14 in the original KO clone, does this reverse the accumulation of acidic vacuoles?

1.2.       Conclusions drawn from data presented in Figure 1 could also be reinforced by modulating T2R14 activity. If we take the data presented at face value, a prediction is that activating T2R14 expressed in OKF6 cells during the starvation period, with a specific agonist, could decrease the number of acidic vacuoles relative to unstimulated cells.

1.3.       Regarding the modulation of autophagy flux and T2R14, data and conclusions are not discussed in the context of T2R14 signalling. Does the data presented in Figure 1 suggest that T2R14 constitutive activity regulates autophagy flux, since there is no agonist treatment? This should be made clear in the text. Can specific T2R14 antagonists act as inverse agonist and reduce the number of acidic vacuoles?

1.4.       Authors should show that the T2R14KO line does not express T2R14.

1.5.       Labelling of graphs in Figure 1 is painfully small on the printed version. Needs to be edited to a larger font.

Point #2

2.1.       In the experiment presented in Figure 2, it seems that the serum starved WT OKF6 cells data is missing. How can one conclude on the effect of reducing expression of ATG7 (knockdown) on autophagy flux without comparing the data to the effect seen in WT cells in the same experiment? The authors suggest a ATG7 independent mechanism…. If one compares the absolute values on the Y axis in Figures 1D and 2D there could actually be a ATG7KD-dependent reduction in acidic vacuoles…

2.2.      Line 246: Fig AI and AIII mean Fig 2B and 2D?

2.3.      Here again, labelling of graphs in Figure 2 is too small.

Point #3

3.1.       Several issues with Figure 3. Labelling is too small, and figure legends are missing in Figures 3B and 3C. Since all the symbols are the same, unlike in Figure 3A, it is impossible to discriminate between the CSP-1, AIP-1, and buffer treatments in Figure 3B and discriminate between CSP-1 and CSP-1 + BAF A1 treatment in Figure 3D.

3.2.      Interpretation of the data is inaccurate. Line 283 and 360-361 state that “…T2R14 -mediated calcium release was not involved in T2R14-dependent autophagy flux in GECs…”. Shouldn’t this read something like this: “…data suggest that autophagy flux does not impact hT2R14-dependent calcium mobilization”, from data in Figure 3C? To propose the conclusion as written, one would have had to monitor autophagy flux in the presence of calcium mobilization pathway inhibitors (PLC inhibitors, IP3R inhibitors, PTx…)…

Point#4

4.1        Several issues with Figure 4. Labelling is too small and Y axis scale should be adjusted to values between 0 and 1 for panels A and B and 0 and 1.5 for panel C.

4.2.      Why use migration of dHL60 cells to monitor the impact of autophagy flux on the secretion of chemoattractants, AMPs, cytokines, and such by GEC cells. Seems like a rather indirect way of answering the question. Direct measurement could be more sensitive and uncover effects. Indeed, treatment with the positive control FMLP, only doubles to migration index of dHL60 cells, as shown in Figure 4C.

Author Response

The paper by Singh et al, entitled “Bitter taste receptor T2R14 and autophagy flux in gingival epithelial cells” is a continuation of an investigation, by the same authors, on the involvement of bitter taste receptors, namely Tas2R14 (T2R14), on gingival epithelial cells-mediated innate immunity (references 10 and 11 in current manuscript draft). In the current study, the authors conduct a series of experiments aimed at uncovering potential T2R14-dependent modulation of autophagy flux in an oral keratinocyte cell line (OKF6), revealing some the molecular/cellular mechanisms governing autophagy in these cells, and the impact on autophagy flux modulation (with the use of pharmacological blockers), and T2R14, on innate immune response, through measurement of migration of neutrophil-like cells, using conditioned media derived from bacteria-infected OKF6 cells.

Unfortunately, the paper in its present form does not address the some of the authors’ questions precisely, and, in some instances, the data interpretation is inconsistent. Furthermore, some conclusions are not fully supported. In one of the last sentences of the introduction (lines 80 and 81), the authors state that they provide evidence that T2R14 modulate autophagic flux in GECs. Not enough data is provided to support this conclusion, in my opinion.

Point #1:

In the first series of experiments, the authors monitor autophagy flux in OKF6 cells, and a derived cell line where T2R14 was knocked out using CRISPR (T2R14KO). Figures 1B and 1D show a modest increase in the numbers of acidic vacuoles in the T2R14KO cells vs WT cells. Transmission electron microscopy data shown in Figures 1F-H seem to confirm this data. However:

1.1.        How can we exclude just a clonal effect of the T2R14KO GECs? The authors do not describe how the knockout cell line was generated but rather point to reference #10 as the source of information. In this refenced paper, CRISPR is used to generate T2R14 knock down OKF6 cells, and single clones are selected. So, what if the data provided in Figure 1 of the current paper is just the result of a specific OKF6 clone, exhibiting more acidic vacuoles upon starvation, while also having no expression of T2R14? If one re-introduces T2R14 in the original KO clone, does this reverse the accumulation of acidic vacuoles?

Response: In our previous work now reference #14 we have described in detailed about the generation of T2R14 KO using CRISPR in OKF6. In total, 30-40 single clones obtained from serial dilutions in 96 plates were screened for mutations in the T2R14 gene using T7E1 assay. We got two biallelic clones (complete knock out of TAS2R14) and out of these two clones one biallelic was used in the current study and in our previous studies. The current clone used in the study has good cell viability and are growing well in normal and serum starved conditions, showing no distorted morphology (Fig S1). We respectfully disagree that the specific OKF6 T2R14 KO cells are exhibiting more acidic vacuoles upon starvation, and we believe the observed effect is because of T2R14 receptor KO.

As the reviewer suggests, we attempted the reintroduction of T2R14 in T2R14KO GEC as described in the methods (lines 120-124) and results (lines 249-252) are presented in fig. S1. Our result suggested T2R14KO GECs transfected with TAS2R14 were less viable (Fig S1) and grow slower than the T2R14KO GEC.. In theory, too many genetic manipulations such as reintroduction of a gene into KO cells causes additional severe stress and the cellular output might not be meaningful or physiological relevant. This we believe might be the case here.  

1.2.       Conclusions drawn from data presented in Figure 1 could also be reinforced by modulating T2R14 activity. If we take the data presented at face value, a prediction is that activating T2R14 expressed in OKF6 cells during the starvation period, with a specific agonist, could decrease the number of acidic vacuoles relative to unstimulated cells.

             Response: Thank you for the suggestions. We used T2R14 agonist diphenhydramine (DPH) which we have used in our previous investigations (3) and its effects have also been confirmed by other investigators (5).We conducted the Acridine orange staining experiment in order to measure the autophagy flux in WT GEC under the serum starved conditions upon treatment with selective agonist DPH. Using 500 µM (EC50) DPH did not lead to any change in cell viability for 24 hours treatment (fig S2). WT GEC upon treatment with T2R14 agonist DPH at 500 µM under serum starved conditions leads to non-significant change in the number of acidic vacuoles relative to the unstimulated cells. This data is now included in the fig. S2. Therefore, our data confirmed that modulating T2R14 activity is not involved in regulation of T2R14 effect in autophagy flux.

1.3.       Regarding the modulation of autophagy flux and T2R14, data and conclusions are not discussed in the context of T2R14 signalling. Does the data presented in Figure 1 suggest that T2R14 constitutive activity regulates autophagy flux, since there is no agonist treatment? This should be made clear in the text. Can specific T2R14 antagonists act as inverse agonist and reduce the number of acidic vacuoles?

             Response: The data presented in figure 1 suggests that the expression of T2R14 is involved in the autophagy flux under serum starved condition. At this point, it is difficult to characterize whether T2R14 constitutive activity is regulating autophagy flux. To prove this hypothesis, one need to use constitutive active mutants of T2R14 or use inverse agonists of T2R14, which are not available. Now in the present study T2R14 antagonist 6-methoxy flavanone (6-MF) was used (PLOS One 2014). This 6-MF was used in our previous publication (3) and by other investigators (5) suggested reduction in T2R14 mediated innate immune responses in GEC and airway cells. Thus, in the current study we tested 6-MF at 30 µM to check its effect on acidic vacuoles formation. Usage of 30 µM 6-MF did not lead to any change in cell viability for 24 hours treatment. Our result suggests that there is no significant change in number of acidic vacuoles compared to the control serum starved cells (fig. S2).

1.4.       Authors should show that the T2R14KO line does not express T2R14.

             Response: This was already published.  (The FASEB Journal. 2021; 35:e21375. https://doi.org/10.1096/fj.202000208R and International Journal of Molecular Sciences. 2021; 22(18):9920. https://doi.org/10.3390/ijms22189920) we have generated T2R14 KO and checked its expression [IJMS 2021 : Supplementary figure S1 and FASEB 2021: figure 3 A and B ).

1.5.       Labelling of graphs in Figure 1 is painfully small on the printed version. Needs to be edited to a larger font. 

 Response: As suggested, Figure 1 is modified to a larger font.

Point #2

2.1.       In the experiment presented in Figure 2, it seems that the serum starved WT OKF6 cells data is missing. How can one conclude on the effect of reducing expression of ATG7 (knockdown) on autophagy flux without comparing the data to the effect seen in WT cells in the same experiment? The authors suggest a ATG7 independent mechanism…. If one compares the absolute values on the Y axis in Figures 1D and 2D there could actually be a ATG7KD-dependent reduction in acidic vacuoles…

            Response: The WT OKF6 and T2R14 KO scramble acridine orange data is included now in Figure 2.

2.2.      Line 246: Fig AI and AIII mean Fig 2B and 2D?

            Response: Thank you. It is corrected now.

2.3.      Here again, labelling of graphs in Figure 2 is too small.

            Response: Labelling size is increased now.

Point #3

3.1.       Several issues with Figure 3. Labelling is too small, and figure legends are missing in Figures 3B and 3C. Since all the symbols are the same, unlike in Figure 3A, it is impossible to discriminate between the CSP-1, AIP-1, and buffer treatments in Figure 3B and discriminate between CSP-1 and CSP-1 + BAF A1 treatment in Figure 3D.

            Response: As suggested, changes are done in Figure 3.

3.2.      Interpretation of the data is inaccurate. Line 283 and 360-361 state that “…T2R14 -mediated calcium release was not involved in T2R14-dependent autophagy flux in GECs…”. Shouldn’t this read something like this: “…data suggest that autophagy flux does not impact hT2R14-dependent calcium mobilization”, from data in Figure 3C? To propose the conclusion as written, one would have had to monitor autophagy flux in the presence of calcium mobilization pathway inhibitors (PLC inhibitors, IP3R inhibitors, PTx…)…

Response: As suggested sentence is now modified in text.   

Point#4

4.1        Several issues with Figure 4. Labelling is too small and Y axis scale should be adjusted to values between 0 and 1 for panels A and B and 0 and 1.5 for panel C.

 Response: Corrections done as suggested.

4.2.      Why use migration of dHL60 cells to monitor the impact of autophagy flux on the secretion of chemoattractants, AMPs, cytokines, and such by GEC cells. Seems like a rather indirect way of answering the question. Direct measurement could be more sensitive and uncover effects. Indeed, treatment with the positive control FMLP, only doubles to migration index of dHL60 cells, as shown in Figure 4C.

 Response: Migration of dHL60 cells is a standard assay for monitoring the neutrophil migration. HL-60 cells are a long established and relatively easy to use cell line, whereas access to primary human neutrophils is too expensive or difficult for performing neutrophil migration assay (6). Direct measurement of neutrophil cells is not feasible and beyond the scope of the study.

References

  1. Singh N, Ghavami S, Chelikani P. Characterization Of Bitter Taste Receptor Dependent Autophagy in Oral Epithelial Cells. bioRxiv. 2024:2024.02.02.578576.
  2. Zhou YW, Sun J, Wang Y, Chen CP, Tao T, Ma M, et al. Tas2R activation relaxes airway smooth muscle by release of Gα. Proc Natl Acad Sci U S A. 2022;119(26):e2121513119.
  3. Medapati MR, Singh N, Bhagirath AY, Duan K, Triggs-Raine B, Batista EL, Jr., et al. Bitter taste receptor T2R14 detects quorum sensing molecules from cariogenic Streptococcus mutans and mediates innate immune responses in gingival epithelial cells. FASEB J. 2021;35(3):e21375.
  4. Medapati MR, Bhagirath AY, Singh N, Schroth RJ, Bhullar RP, Duan K, et al. Bitter Taste Receptor T2R14 Modulates Gram-Positive Bacterial Internalization and Survival in Gingival Epithelial Cells. Int J Mol Sci. 2021;22(18).
  5. Carey RM, Palmer JN, Adappa ND, Lee RJ. Loss of CFTR function is associated with reduced bitter taste receptor-stimulated nitric oxide innate immune responses in nasal epithelial cells and macrophages. Front Immunol. 2023;14:1096242.
  6. Babatunde KA, Wang X, Hopke A, Lannes N, Mantel PY, Irimia D. Chemotaxis and swarming in differentiated HL-60 neutrophil-like cells. Sci Rep. 2021;11(1):778.

Round 2

Reviewer 1 Report

Comments and Suggestions for Authors

According to responses to my comment, manuscript has been improved and can be now published.

Reviewer 2 Report

Comments and Suggestions for Authors

Paper accepted in revised form.